

# N-acetylcysteine protects against motor, optomotor and morphological deficits induced by 6-OHDA in zebrafish larvae

Radharani Benvenutti[1], Matheus Marcon[1], Carlos G. Reis[1], Laura R. Nery[2], Camila Miguel[3], Ana P. Herrmann[4], Monica R.M. Vianna[2,3] and Angelo Piato[1,4]

[1] Programa de Pós-graduação em Neurociências, Universidade Federal do Rio Grande do Sul, Porto Alegre, Rio Grande do Sul, Brazil
[2] Programa de Pós-graduação em Biologia Celular e Molecular, Pontifícia Universidade Católica do Rio Grande do Sul, Porto Alegre, Rio Grande do Sul, Brazil
[3] Programa de Pós-graduação em Zoologia, Pontifícia Universidade Católica do Rio Grande do Sul, Porto Alegre, Rio Grande do Sul, Brazil
[4] Programa de Pós-graduação em Farmacologia e Terapêutica, Universidade Federal do Rio Grande do Sul, Porto Alegre, Rio Grande do Sul, Brazil

Corresponding author
Angelo Piato, angelopiato@ufrgs.br

## ABSTRACT

**Background:** Parkinson's disease (PD) is the second most common neurodegenerative disorder. In addition to its highly debilitating motor symptoms, non-motor symptoms may precede their motor counterparts by many years, which may characterize a prodromal phase of PD. A potential pharmacological strategy is to introduce neuroprotective agents at an earlier stage in order to prevent further neuronal death. N-acetylcysteine (NAC) has been used against paracetamol overdose hepatotoxicity by restoring hepatic concentrations of glutathione (GSH), and as a mucolytic in chronic obstructive pulmonary disease by reducing disulfide bonds in mucoproteins. It has been shown to be safe for humans at high doses. More recently, several studies have evidenced that NAC has a multifaceted mechanism of action, presenting indirect antioxidant effect by acting as a GSH precursor, besides its anti-inflammatory and neurotrophic effects. Moreover, NAC modulates glutamate release through activation of the cystine-glutamate antiporter in extra-synaptic astrocytes. Its therapeutic benefits have been demonstrated in clinical trials for several neuropsychiatric conditions but has not been tested in PD models yet.
**Methods:** In this study, we evaluated the potential of NAC to prevent the damage induced by 6-hydroxydopamine (6-OHDA) on motor, optomotor and morphological parameters in a PD model in larval zebrafish.
**Results:** NAC was able to prevent the motor deficits (total distance, mean speed, maximum acceleration, absolute turn angle and immobility time), optomotor response impairment and morphological alterations (total length and head length) caused by exposure to 6-OHDA, which reinforce and broaden the relevance of its neuroprotective effects.
**Discussion:** NAC acts in different targets relevant to PD pathophysiology. Further studies and clinical trials are needed to assess this agent as a candidate for prevention and adjunctive treatment of PD.

## INTRODUCTION

Parkinson's disease (PD) is the second most common neurodegenerative disorder in the world, affecting on average 2–3% of the individuals older than 65 years (*Poewe et al., 2017*). This condition originates from progressive death of dopamine (DA) neurons in the substantia nigra *pars compacta* of the midbrain and is characterized by motor and non-motor symptoms (*Kalia & Lang, 2015*). Although the etiology of this multifactorial disease remains unknown, studies demonstrate a key role of oxidative stress in the development of PD in addition to mitochondrial dysfunction, neuroinflammation, dysfunction of the ubiquitin-proteasome system and Lewy body formation (*Ryan et al., 2015*; *Blesa et al., 2015*). 6-Hydroxydopamine (6-OHDA) is a neurotoxin that has been widely used in animal models to mimic pathogenic events and behavioral features observed in PD (*Anichtchik et al., 2004*; *Blandini & Armentero, 2012*; *Feng et al., 2014*; *Zhang et al., 2017*). 6-OHDA is a reactive structural analogue of DA, uptaken into the neuron by the DA transporter once it crosses the blood-brain barrier. 6-OHDA inhibits the mitochondrial complex I, resulting in an increase of reactive oxygen species production and impairment of the ATP generation, which leads to dopaminergic neuron death (*Jackson-Lewis, Blesa & Przedborski, 2012*).

The available drugs for the treatment of PD, such as L-DOPA, are focused on increasing dopaminergic neurotransmission. However, besides inducing several adverse effects and lacking efficacy in the treatment of non-motor symptoms, none of these drugs are able to cure the disease or even slow the progression of neuronal loss (*Poewe et al., 2017*). A possible explanation for the relative inefficacy of such treatments is related to the tardiness of their onset, which usually begins upon the appearance of motor symptoms, when neuronal death is already at an advanced stage (*Berg et al., 2012*). Therefore, a potential pharmacological strategy would be to identify individuals in the prodromal phase and to introduce neuroprotective agents at an earlier stage in order to prevent further neuronal death (*Dexter & Jenner, 2013*).

Drug repurposing is a term used to describe the repositioning of known compounds, which are already marketed, to target novel therapeutic purposes. It is an attractive strategy for drug development due to the savings in research, funding and time (*Insel et al., 2013*; *Langedijk et al., 2015*; *Klug, Gelb & Pollastri, 2016*). In this context, *N*-acetylcysteine (NAC) may be a potential candidate for drug repurposing. NAC has been used against paracetamol overdose hepatotoxicity by restoring hepatic concentrations of glutathione (gamma-glutamylcysteinylglycine; GSH), and as a mucolytic in chronic obstructive pulmonary disease by reducing disulfide bonds in mucoproteins. Moreover, even at high doses, NAC appears to be safe in humans (*Whyte, Francis & Dawson, 2007*). More recently, several studies have evidenced that NAC has a multifaceted mechanism of action, acting as an indirect antioxidant for being a GSH precursor, and showing anti-inflammatory and neurotrophic activities. Both animal and human studies have demonstrated that NAC is able to increase neuronal levels of GSH

(*Tchantchou et al., 2005*; *Clark et al., 2010*; *Holmay et al., 2013*). Moreover, NAC modulates glutamate release through activation of the cystine-glutamate antiporter in extra-synaptic astrocytes (*Berk et al., 2013*). Furthermore, NAC indirectly regulates NMDA receptor activity, since GSH binds to a redox sensitive site on NMDA receptor and modulates its activity (*Steullet et al., 2006*). Therapeutic benefits of NAC have been demonstrated in clinical trials for several neuropsychiatric conditions (*Dean, Giorlando & Berk, 2011*).

In recent years, the zebrafish has become a powerful tool to investigate and develop new drugs in neurological and neuropsychiatric research (*MacRae & Peterson, 2015*; *Fontana et al., 2018*). Despite its reduced size and complexity, zebrafish brains have neuroanatomical areas homologous to mammals, including the striatum, therefore it has also been used and standardized in PD studies as a model for drug screening and investigation of pathophysiology (*Rink & Wullimann, 2004*; *Xi, Noble & Ekker, 2011*). In addition to the models based on genetic manipulation, several studies have used neurotoxins such as 1-methyl-4-phenyl-1,2,3,6-tetrahydropyridine (MPTP) and 6-OHDA to model PD in zebrafish (*McKinley et al., 2005*; *Zhang et al., 2012*, *2017*; *Chong et al., 2013*; *Feng et al., 2014*). A remarkable advantage of the zebrafish PD model is that the blood-brain barrier in the larval stage is more permeable to neurotoxins such as MPTP and 6-OHDA as compared to rodents (*Feng et al., 2014*) (*Jackson-Lewis, Blesa & Przedborski, 2012*).

Because of its multifaceted mechanism of action, we hypothesized that NAC may have a neuroprotective effect and prevent or minimize motor signs related to PD. To address this question, we evaluated the potential of NAC to prevent the injury caused by 6-OHDA exposure on motor, optomotor response and morphological parameters in zebrafish larvae.

## MATERIALS AND METHODS

### Chemicals and reagents

*N*-acetylcysteine, 6-OHDA hydrobromide 95%, methyl cellulose and ethyl 3-aminobenzoate methanesulfonate (MS-222) were purchased from Sigma-Aldrich (St. Louis, MO, USA).

### Animals

Embryos and larvae (0 and 7 days post-fertilization (dpf)) of AB strain zebrafish (*Danio rerio*) were used. The animals were obtained from our breeding colony, which was maintained in recirculating systems (Zebtec, Tecniplast, Italy) with reverse osmosis filtered water equilibrated to reach the species standard parameters including temperature (28 ± 2 °C), pH (7 ± 0.5), conductivity and ammonia, nitrite, nitrate and chloride levels. Water used in the experiments was obtained from a reverse osmosis apparatus (18 MOhm/cm) and was reconstituted with marine salt (Crystal Sea™, Marinemix, Baltimore, MD, USA) at 0.4 ppt. The total organic carbon concentration was 0.33 mg/L. The total alkalinity (as carbonate ion) was 0.030 mEq/L. The animals were kept with a light/dark cycle of 14/10 h. Larvae from the developmental stages used in this study rely on the yolk sac for nutrition and feeding the animals is not necessary. At the end of the

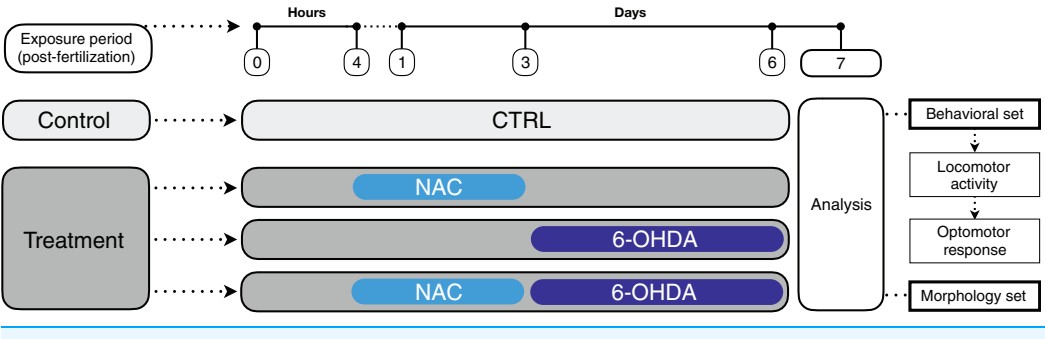

**Figure 1 Timeline representing the experimental design and treatment groups.**

experiment, the larvae were euthanized by hypothermia. All protocols were approved by the Animal Care Committee of Pontifícia Universidade Católica do Rio Grande do Sul (#7994/17).

## Experimental design

The experiments were performed according to Fig. 1. In a breeding tank, females and males (1:2) were separated overnight by a transparent barrier, which was removed after the lights went on in the following morning. The fertilized eggs that were retained in the bottom of the fitted tank were collected, washed and gently placed in a 6-well plate (15 animals per well) or 24-well plate (four animals per well), according to the treatment groups and experiment (6-well plate for behavioral tests and 24-well plate for morphological analysis). We chose not to use 96-well plates to avoid the bias of isolating the larvae, solution evaporation and skeletal malformations due to decreased swimming space (see review by *Beekhuijzen et al., 2015*). The volume used in the 6-well and 24-well plates was 5 and 2 ml per well, respectively.

A set of animals ($n = 8–11$) was used for the behavioral tests (locomotor activity first, followed by optomotor response test) and another set was used for the morphological analyses ($n = 11–12$). Animals were randomly assigned to the experimental groups following simple randomization procedures without stratification (computerized random numbers). All experimenters who analyzed the images and videos were blind to treatment—each experimental group was given a code that was revealed only after the analyses were performed (codification was carried out by a researcher who did not participate in the analyses). Total distance traveled in the locomotor activity test was considered our primary outcome, while the remaining measures were secondary outcomes.

At 4 hours post-fertilization (hpf) (*Beekhuijzen et al., 2015*) the embryos were exposed to 1 mg/L NAC or system water. The medium was not changed until 3 dpf. NAC concentration was determined in a pilot study as well as in our previous studies (*Mocelin et al., 2015*, *2017*).

At 3, 4, 5 and 6 dpf the animal medium was changed for 250 μM 6-OHDA solution or system water (*Zhang et al., 2017*). The NAC and 6-OHDA solutions were diluted in system water in a light-protected environment immediately before use. To avoid

interference in the solution pH, we did not use ascorbic acid as a 6-OHDA conservator. For this reason, we opted to replace 6-OHDA solution daily.

At 7 dpf, the motor and morphological parameters of the larvae were analyzed. Embryos and larvae had their mortality and morphology observed daily under the stereomicroscope. There was no difference in the mortality rate between the experimental groups. When applicable, dead animals and the corium were removed (death events occurred before 6-OHDA exposure and were not associated with NAC treatment). The experiments were run twice to confirm the data (no differences were observed between the cohorts).

### Locomotor behavior

At 7 dpf the larvae were transferred to the experimental room within the animal facility and individually placed in a 24-well plate filled with 2 ml of system water. The locomotor activity was recorded and analyzed during 5 min, following 1 min of acclimation, using Noldus Ethovision XT system (Wageningen, Netherlands). Total distance, mean speed, maximum acceleration, absolute turn angle and immobility time were considered the main parameters of locomotor activity (*Altenhofen et al., 2017*; *Nery et al., 2017*). The experiments were performed in a temperature-controlled room (27 ± 2 °C) between 9:00 and 14:00 h.

### Optomotor response

The optomotor response test allows the assessment of an innate response behavior, indirectly related to cognition (*Nery et al., 2017*), described as the swimming of zebrafish larvae in the same direction of a moving pattern of stripes (*Fleisch & Neuhauss, 2006*). We performed this test adapted from *Creton (2009)*. The apparatus consists of a Petri dish positioned over a LCD screen. After being tested for locomotor behavior, the larvae were placed in groups of 10 on the Petri dish filled with 5 ml of system water. After 2 min of acclimation in which the screen was white, the larvae were exposed to a visual stimulus consisting of a moving pattern of red and white stripes (24.5 cm wide and 1.5 cm high). First, the stripes move up at 1 cm/s for 1 min, and then move down at 1 cm/s for 1 min. This pattern repeated four times, with a 5 s interval between each 1 min show, in which the screen went white. The entire experiment was recorded and then analyzed by investigators who were blind to the experimental groups. For the data analysis, the Petri dish was virtually divided into two halves (upper and lower) and the number of animals in the stimulus zone (the region towards which the pattern moved) was counted during the 5 s interval of white screen.

### Morphological analysis

At 7 dpf the larvae were individually placed in a Petri plate containing 200 μL of 3% methylcellulose with 0.1 g/L ethyl 3-aminobenzoate methanesulfonate (MS-222) solution and each larva was photographed using an inverted stereomicroscope (Nikon, Melville, NY, USA) connected to NIS-Elements Viewer software. Total length, head length, forebrain width, midbrain width and eyes distance were considered the main

morphological parameters (*Altenhofen et al., 2017*) and measured by investigators who were blind to the experimental groups using Image J software. The experiments were performed in a temperature-controlled room (27 ± 2 °C) between 13:00 and 15:00 h.

## Statistical analysis

Data were analyzed after normality and homogeneity of variance (D'Agostino-Person and Levene tests, respectively) confirmation using two-way ANOVA (type III sums of squares) to identify the main motor and morphological effects of pretreatment (NAC exposure or not) and treatment (6-OHDA exposure or not) and their interaction, followed by Bonferroni post hoc test. Sample sizes were calculate a priori based on data from the literature and pilot experiments and the observed power for our primary endpoint (total distance traveled) was 75% for NAC factor, 65% for 6-OHDA factor, and 92% for the interaction. Outliers were removed according to Tukey's boxplot method. Data were expressed as the mean ± S.E.M. For all comparisons, the significance level was set at $p < 0.05$.

## RESULTS

Figure 2 shows the effect of NAC (1 mg/L) on locomotor behavior in 7 dpf larvae exposed to 6-OHDA (250 μM). 6-OHDA caused a decrease in total distance (Fig. 2A), mean speed (Fig. 2B) and maximum acceleration (Fig. 2C), while it increased absolute turn angle (Fig. 2D) and immobility time (Fig. 2E). In all locomotor parameters, NAC was able to prevent the locomotor deficits induced by 6-OHDA. NAC per se did not present statistical differences when compared to the control group. Table 1 summarizes the two-way ANOVA analysis.

Figure 3 shows the effect of NAC (1 mg/L) on the optomotor response test in 7 dpf larvae exposed to 6-OHDA. Larvae exposed to 6-OHDA spent less time in the stimulus zone, and NAC was able to prevent this optomotor deficit. NAC per se did not alter this parameter.

Figure 4 shows the effect of NAC (1 mg/L) on morphological parameters in 7 dpf larvae exposed to 6-OHDA (250 μM). 6-OHDA decreased the total length (Fig. 4A) and head length (Fig. 4B), whereas treatment with NAC prevented this effect. NAC per se did not induce morphological alterations. There was no statistical difference in any experimental groups regarding forebrain width (Fig. 4C), midbrain width (Fig. 4D) and eyes distance (Fig. 4E). Table 2 summarizes the two-way ANOVA analysis.

## DISCUSSION

Our results demonstrated that 6-OHDA at 250 μM is able to induce motor and optomotor deficits and morphological alterations in zebrafish larvae at 7 dpf. Interestingly, NAC (1 mg/L) prevented these effects when administered at the very onset of zebrafish embryos development (4 hpf), showing a clear neuroprotective effect against the neurotoxin.

In the field of animal models of PD induced by neurotoxins such 6-OHDA, the use of zebrafish has increased, probably due to its various benefits when compared to mammal

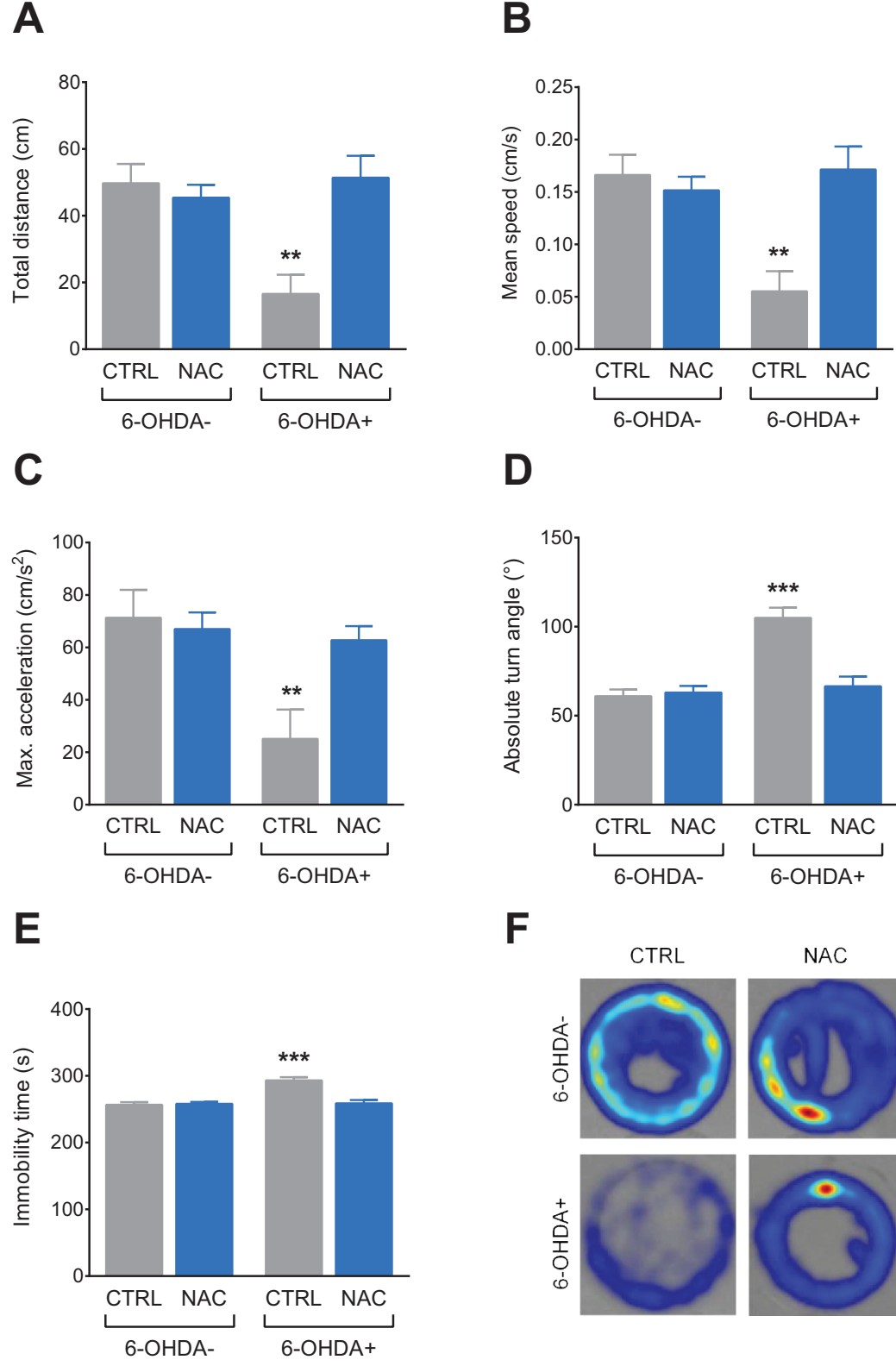
**Figure 2 Effects of NAC on 6-OHDA-induced locomotor behavior deficits in zebrafish larvae.** (A) Total distance travelled, (B) mean speed, (C) maximum acceleration, (D) absolute turn angle, (E) immobility time and (F) representative color heatmap of the behavior of one larvae from each treatment group during the trial 5 min duration. Data are expressed as mean ± standard error of mean (S.E.M.). $n = 8$–11. Two-way ANOVA followed by Bonferroni post hoc test. Study sites: CTRL, control; NAC, N-acetylcysteine; 6-OHDA, 6-hydroxydopamine. **$p < 0.01$, ***$p < 0.001$ vs. control group (6-OHDA-).

**Table 1 The main effects of behavioral analysis and the interaction between pretreatment with NAC and treatment with 6-OHDA.**

| Dependent variable | Effects | F-value | DF | P-value |
|---|---|---|---|---|
| Total distance | Interaction | 12.10 | 1.35 | **0.0014** |
| | 6-OHDA | 5.81 | 1.35 | **0.0212** |
| | NAC | 7.31 | 1.35 | **0.0105** |
| Mean speed | Interaction | 12.13 | 1.35 | **0.0014** |
| | 6-OHDA | 5.88 | 1.35 | **0.0206** |
| | NAC | 7.29 | 1.35 | **0.0106** |
| Maximum acceleration | Interaction | 6.46 | 1.35 | **0.0156** |
| | 6-OHDA | 9.36 | 1.35 | **0.0042** |
| | NAC | 4.09 | 1.35 | 0.0506 |
| Absolute turn angle | Interaction | 17.83 | 1.35 | **0.0002** |
| | 6-OHDA | 29.59 | 1.35 | **0.0001** |
| | NAC | 14.45 | 1.35 | **0.0006** |
| Immobility time | Interaction | 17.62 | 1.35 | **0.0002** |
| | 6-OHDA | 19.06 | 1.35 | **0.0001** |
| | NAC | 14.94 | 1.35 | **0.0005** |
| Optomotor response | Interaction | 55.84 | 1.32 | **0.0001** |
| | 6-OHDA | 74.69 | 1.32 | **0.0001** |
| | NAC | 49.05 | 1.32 | **0.0001** |

Notes:
DF, degrees of freedom.
Significant effects ($p < 0.05$) are given in bold font.

models (*Babin, Goizet & Raldúa, 2014*; *MacRae & Peterson, 2015*). We did not evaluate the 6-OHDA-induced damage to dopaminergic neurons ourselves, however we followed the protocol described by *Zhang et al. (2017)* who evaluated the DA neuron system of zebrafish larvae by immunofluorescent staining with a specific antibody against anti-TH and found that treatment with 6-OHDA decreased the number of DA neurons markedly in the diencephalon of larvae zebrafish. By causing the death of dopaminergic neurons from important pathways associated with movement regulation, it has been shown in several studies that 6-OHDA is able to cause locomotor deficits in zebrafish larvae (*Feng et al., 2014*; *Zhang et al., 2017*) and adults (*Anichtchik et al., 2004*). According to what is shown in the literature, our results demonstrate that 6-OHDA caused locomotor deficit in all analyzed parameters, causing decrease in distance, mean speed and maximum acceleration and increase in absolute turn angle and immobility time. Our findings show,

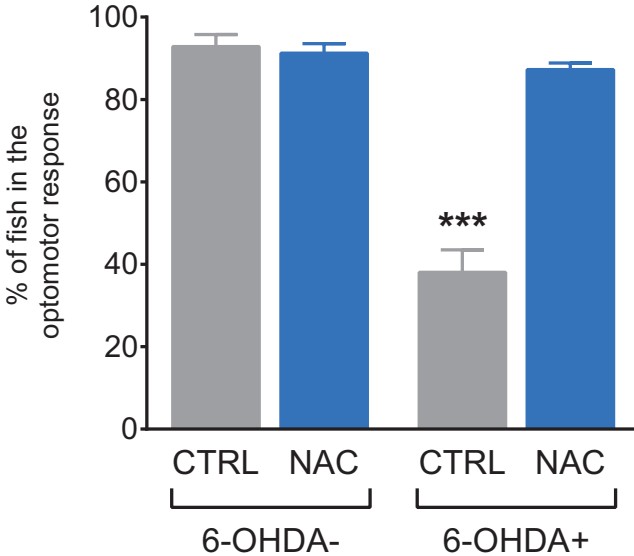

**Figure 3 Effects of NAC on 6-OHDA-induced optomotor response deficit in zebrafish larvae.** Data are expressed as mean ± standard error of mean (S.E.M.). *n* = 10. Two-way ANOVA followed by Bonferroni post hoc test. Study sites: CTRL, control; NAC, *N*-acetylcysteine; 6-OHDA, 6-hydroxydopamine. ***$p < 0.001$ vs. control group (6-OHDA-).

for the first time, that pre-treatment with NAC is capable of preventing the locomotor deficits induced by 6-OHDA. In another study, NAC improved the behavioral damages and dopaminergic neurons loss induced by rotenone in an animal model of PD in rats, which is in accordance with our results, further suggesting the neuroprotective effects of NAC in behavioral and neurochemical parameters (*Rahimmi et al., 2015*).

The optomotor response test evaluates zebrafish's sensory performance, in addition to its responsiveness to the environment (*Maaswinkel & Li, 2003*; *Creton, 2009*). Several PD patients demonstrate sensory dysfunctions, such as changes in visual perception, including color perception and contrast sensitivity (*Weil et al., 2016*). These patients have difficulty performing complex visual tasks such as mental rotation and emotion recognition (*Weil et al., 2016*). As sensory impairment is a non-motor PD symptom, it is important to evaluate the effects of NAC on optomotor parameters. Thus, our data demonstrated for the first time that larvae exposed to 6-OHDA spent less time in the stimulus zone, presenting an optomotor damage. NAC was able to prevent the deficit in optomotor response in larvae exposed to 6-OHDA, demonstrating the neuroprotective role of NAC in pathways related to sensory functions of zebrafish larvae. Although there is a motor component to this test, we believe the deficits on locomotor activity alone would not explain the poor performance in this test, and sensory impairment is likely to be involved as well.

Studies using tyrosine hydroxylase (TH) and DA transporter staining showed that after 24 hpf the zebrafish larvae already have functional dopaminergic neurons in areas such as posterior tuberculum in the midbrain and within 4 dpf all dopaminergic pathways are present (*Nellore & Nandita, 2015*). Our study also shows that 6-OHDA caused

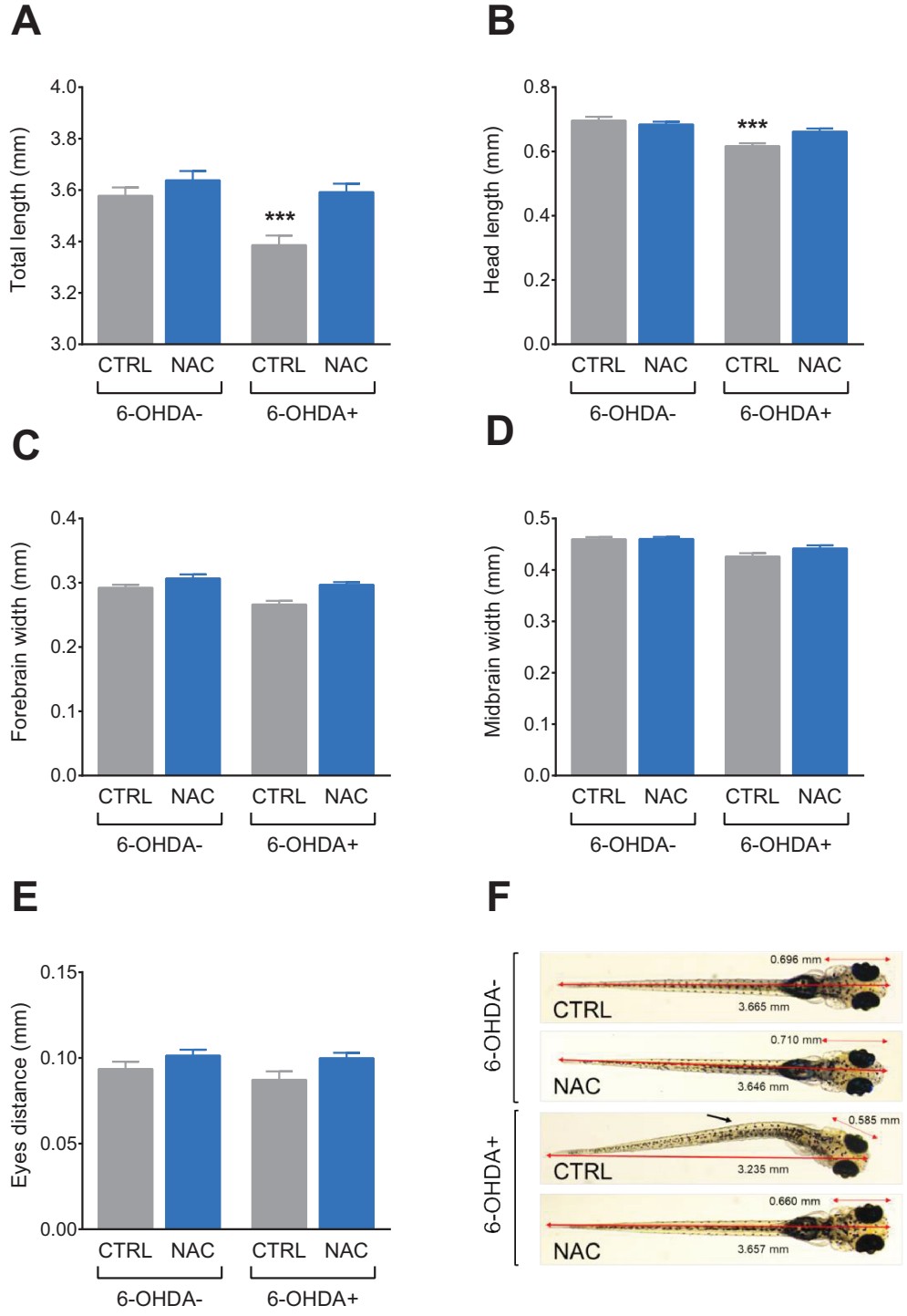

**Figure 4 Effects of NAC on 6-OHDA-induced morphological alterations in larvae zebrafish.** (A) Total length, (B) head length, (C) forebrain width, (D) midbrain width, (E) eyes distance and (F) pictures of a larva from each treatment group. The arrow indicates a notable morphological alteration in the shape of the larvae. Data are expressed as mean ± standard error of mean (S.E.M.). $n$ = 11–12. Two-way ANOVA followed by Bonferroni post hoc test. Study sites: CTRL, control; NAC, $N$-acetylcysteine, 1 mg/L; 6-OHDA, 6-hydroxydopamine, 250 µM. ***$p$ < 0.001 vs. control group (6-OHDA-). Image credit/source: (F) Radharani Benvenutti.

**Table 2 The main effects of morphological analysis and the interaction between pretreatment with NAC and treatment with 6-OHDA.**

| Dependent variable | Effects | F-value | DF | P-value |
| --- | --- | --- | --- | --- |
| Total length | Interaction | 4.42 | 1.41 | **0.0417** |
|  | 6-OHDA | 11.77 | 1.41 | **0.0014** |
|  | NAC | 14.72 | 1.41 | **0.0004** |
| Head length | Interaction | 8.56 | 1.41 | **0.0056** |
|  | 6-OHDA | 27.28 | 1.41 | **0.0001** |
|  | NAC | 2.87 | 1.41 | 0.0974 |
| Forebrain width | Interaction | 2.40 | 1.41 | 0.1284 |
|  | 6-OHDA | 11.87 | 1.41 | **0.0013** |
|  | NAC | 18.66 | 1.41 | **0.0001** |
| Midbrain width | Interaction | 2.07 | 1.41 | 0.1578 |
|  | 6-OHDA | 24.72 | 1.41 | **0.0001** |
|  | NAC | 2.36 | 1.41 | 0.1318 |
| Eyes distance | Interaction | 0.35 | 1.41 | 0.5534 |
|  | 6-OHDA | 0.96 | 1.41 | 0.3306 |
|  | NAC | 6.58 | 1.41 | **0.0140** |

Notes:
DF, degrees of freedom.
Significant effects ($p < 0.05$) are given in bold font.

morphological changes in zebrafish larvae, producing apparent body curvature and decrease of total length and head length. Thereby, we can assume that the pathways that suffered neuronal death play a key role in regulating the development of the zebrafish larvae. Similar morphological observations were seen in zebrafish larvae exposed to pesticides such as paraquat and rotenone (*Bretaud, Lee & Guo, 2004*). In the treatment groups that received NAC, no morphological changes were observed, even when exposed to 6-OHDA. Because of its mechanism of action, NAC is likely to have protected these pathways, preventing the morphological changes caused by 6-OHDA. NAC (100 mg/kg) was able to increase the levels of the dopaminergic marker TH in the striatum of mice exposed to 6-OHDA, reinforcing the important effect of NAC in the preservation of the dopaminergic system (*Nouraei et al., 2016*). We observed main effects of NAC in the direction of increased forebrain width and (consequently) eye distance (Table 2)—this is very different from the damaging effects of 6-OHDA and other toxins, which are expected to decrease such values. We speculate that the antioxidant and neurotrophic properties of NAC may have boosted larvae development.

Several studies have shown that NAC has an indirect antioxidant effect through its ability to provide cysteine for the synthesis of GSH, which leads to an increase of neuronal levels of GSH, both in animal models (*Tchantchou et al., 2005*; *Clark et al., 2010*) and in humans (*Holmay et al., 2013*). However, as demonstrated by *Misra (1974)*, thiols can undergo autoxidation in solution forming superoxide free radical and thyl, so it is possible that over the course of days, a portion of NAC can undergo autoxidation, generating these oxygen radicals. These oxygen radicals, in turn, would be sources of damage and

would stimulate transcription factors like Nuclear factor (erythroid-derived 2)-like 2 to regulate the expression of antioxidant defenses such as GSH. We believe that the neuroprotective effect of NAC found in this study may be a result of these two events. In addition, NAC has anti-inflammatory properties by reducing pro-inflammatory cytokine levels, including interleukin (IL)-6, IL-1β and tumor necrosis factor alpha (*Dean, Giorlando & Berk, 2011*). Considering that neuroinflammation mediates an important key role in neurodegeneration associated with PD, studies have shown the preventive potential of non-steroidal anti-inflammatory drugs in modifying some pathophysiological aspects of PD, indicating drugs with an anti-inflammatory profile could act as neuroprotectors (*Rees et al., 2011*). NAC also shows glutamatergic modulator activities by regulating neuronal exchange of glutamate through the cystine-glutamate antiporter, in addition to regulating the dopaminergic transmission (*Dean, Giorlando & Berk, 2011*; *Berk et al., 2013*). NAC also reduces oxidative damage markers (*Alboni et al., 2013*), increases the number of brain synapses (*Samuni et al., 2013*) and activates the mitochondrial complex I (*Samuni et al., 2013*). Although there is limited research about NAC in Parkinson's disease, some studies have demonstrated that NAC has potential as a therapeutic strategy for PD prevention and treatment in humans and PD animal models (*Muñoz et al., 2004*; *Martínez-Banaclocha, 2012*; *Katz et al., 2015*; *Nouraei et al., 2016*; *Coles et al., 2017*).

Studies have indicated NAC effects as an anxiolytic (*Mocelin et al., 2015*; *Santos et al., 2017*) and antidepressant drug (*Magalhães et al., 2011*; *Berk et al., 2013*; *Costa-Campos et al., 2013*; *Pilz et al., 2015*). Therefore, NAC has advantages over the existing therapies since, besides having a potential preventive effect, it may be able to treat the non-motor aspects of PD, which include depression, anxiety, cognitive impairment and sleep disturbances (*Klockgether, 2004*).

It is increasingly necessary to develop drugs with a multifaceted mechanism capable of acting on several targets that are altered in neurodegenerative diseases, such as PD, more effectively and with fewer adverse effects. Translational research aims to serve as a powerful tool in the development of pharmacological interventions that fulfill this goal (*Pickart & Klee, 2014*). Our results propose the investigation of the effect of NAC in the prevention and treatment of PD, showing a protective effect of NAC in behavioral and morphological parameters in a translational model of PD in zebrafish. However, further research is needed to evaluate the action of NAC in other important PD markers. Therefore, we intend to perform additional studies to analyze the role of NAC in markers of oxidative stress, apoptosis and TH in the present model of PD in zebrafish.

## CONCLUSION

This study demonstrated that NAC was able to prevent the behavioral deficits and morphological alterations induced by 6-OHDA, which supports its neuroprotective effect. For having differentiated profile and mechanism of action, NAC is a candidate for the prevention and treatment of PD, acting in different aspects of the disease.

### Funding

This work was supported by Fundação de Amparo à Pesquisa do Estado do Rio Grande do Sul (FAPERGS) and by the Conselho Nacional de Desenvolvimento Científico e Tecnológico (CNPq, #401162/2016-8 and #302800/2017-4). The funders had no role in study design, data collection and analysis, decision to publish, or preparation of the manuscript.

### Grant Disclosures

The following grant information was disclosed by the authors:
Fundação de Amparo à Pesquisa do Estado do Rio Grande do Sul.
Conselho Nacional de Desenvolvimento Científico e Tecnológico: #401162/2016-8 and #302800/2017-4.

### Competing Interests

Angelo Piato is an Academic Editor for PeerJ.

### Author Contributions

- Radharani Benvenutti conceived and designed the experiments, performed the experiments, analyzed the data, prepared figures and/or tables, authored or reviewed drafts of the paper, approved the final draft.
- Matheus Marcon performed the experiments, prepared figures and/or tables, authored or reviewed drafts of the paper, approved the final draft.
- Carlos G. Reis performed the experiments, authored or reviewed drafts of the paper, approved the final draft.
- Laura R. Nery performed the experiments, authored or reviewed drafts of the paper, approved the final draft.
- Camila Miguel performed the experiments, authored or reviewed drafts of the paper, approved the final draft.
- Ana P. Herrmann conceived and designed the experiments, analyzed the data, prepared figures and/or tables, authored or reviewed drafts of the paper, approved the final draft.
- Monica R.M. Vianna conceived and designed the experiments, analyzed the data, contributed reagents/materials/analysis tools, authored or reviewed drafts of the paper, approved the final draft.
- Angelo Piato conceived and designed the experiments, analyzed the data, contributed reagents/materials/analysis tools, prepared figures and/or tables, authored or reviewed drafts of the paper, approved the final draft.

### Animal Ethics

The following information was supplied relating to ethical approvals (i.e., approving body and any reference numbers):

All protocols were approved by the Animal Care Committee of Pontifícia Universidade Católica do Rio Grande do Sul (#7994/17).

## Data Availability

The raw data are provided in a Supplemental File.

## Supplemental Information

Supplemental information for this article can be found online at http://dx.doi.org/10.7717/peerj.4957#supplemental-information.

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
