# Peer review of "N-acetylcysteine protects against motor, optomotor and morphological deficits induced by 6-OHDA in zebrafish larvae"

_PeerJ, doi:10.7717/peerj.4957_

## Round 0.1 · original submission · Major Revisions

Dear Dr. Angelo,

Your manuscript has been reviewed by one independent reviewer and by myself (see below). As you can see, the reviewer has raised several points that are critical to be considered regarding the methodological approaches used in the paper (particularly from the point of view of statistical design to be used to interpret the data). I have also raised some points regarding the necessity of further experiment to increase the strength of your study. If you are willing to revise the manuscript, please, to facilitate the re-evaluation of the manuscript, describe clearly in the rebuttal letter where you have introduced the changes to attend the points raised by the reviewer and by myself. If you do not agree with specific comments, please, give a clear and rational rebuttal to them. If you have any additional questions, please, let me know. I have listed my comments below:

This is a straightforward but interesting study showing the neuroprotective effect of NAC against 6OHDA.

Abstract
Authors have to be more specific and define which morphological changes were changed by NAC.
Furthermore, NAC is, normally, not a direct antioxidant. It is a soft nucleophile that will react rapidly only with soft-electrophiles. The interaction of NAC with hard electrophiles is expected to be very slow. In short, without consulting the rate constants of NAC with the potential electrophiles generated by 6OHDA authors have to be much more conservative throughout the paper an clearly state that NAC is possibly acting as an indirect antioxidant that stimulates the synthesis of GSH. GSH by its turn can be a powerful antioxidant as a substrate of several antioxidant enzymes.

In abstract. as well in the text, author have to emphasize that NAC was originally (and still is) used to increase the GSH in the liver of patients overdosed with paracetamol. Author has also to be clarified whether or not the mucolytic and the other protective effects in lung diseases are mediated by an increase in GSH. Furthermore, authors have to cite that even at high doses it appears to be safe in humans (it was used at megadoses in COPOD).

iNTRODUCTION
Instead of emphasizing the PD diseases in the introduction, authors have to briefly cite that dopaminergic neuron damage is involved in the PD and that 6OHDA can be used to investigate some mechanisms involved in dopaminergic demise in PD. It is hard to believe that an acute exposure to DA neurons damaging agent will simulate the PD.
Indeed, the mechanisms of 6OHDA have to be cited and, particularly, whether or not NAC can increase the levels of GSH in neuronal cells (either in vitro or in vivo models of neurodegeneration).
The mechanisms involved in the modulation of the glutamatergic system by NAC have also to be clarified. Does NAC the glutamatergic system by increasing the synthesis of GSH or by modulating the redox state of glutamatergic receptors (for instance, the NMDA)?


MATERIAL AND METHODS
The volume of the wells, I mean, the volume used in the wells (6 and 24) have to be cited.

The authors have to determine the concentration of NAC along of the time. This can be easily done using the DTNB or the Ellman's reagent. Thiols can be auto-oxidized and even generate thiyl and oxygen radical (see, for instance, the classical paper of Misra, Journal of Biological Chemistry. 1974 Apr 10;249(7):2151-5). The protective effect observed can either be determined by a low level of auto-oxidation of NAC and pro-oxidizing intermediates that have boosted the antioxidant defenses of larvae.
Furthermore, authors have to do additional experiments to test the effects of cysteine and GSH (also their auto-oxidation in the medium). These experiments are critical to determining whether or not any -SH containing molecule can afford protection against 6OHDA in larvae of fish. If the modulation of the glutamatergic system by NAC cited by the authors in the introduction of the manuscript is mediated by NMDA redox modulation, authors have also to include DTT in their experiments.

Ideally, authors should also follow the oxidation of 6OHDA along the day. Indeed, only with this they can guarantee that the effects were mediated by the 6OHDA itself or by reactive species genereted during its auto-oxidation.

I am not sure that calling the optomotor as a "cognitive" evaluation. Indeed, if is an innate response, I mean, if the test does not evaluate any kind of learning and or behavioral change induced by experience, the test cannot be considered as a cognitive one. Furthermore, the validity of this test can be questioned on the basis of the motor impairment found in 6OHDA-exposed larvae. This aspect is critical and author have to be very conservative when they are inferring that NAC prevented the cognitive impairments caused by 6OHDA. Indeed, authors have not demonstrated clearly a cognitive impairment, but a incapacity of larvae to have normal motor activity.

Without determining the effect of 6OHDA in the dopaminergic neurons, authors can not infer about the participation of them in the gross morphological alterations observed.

The citation that NAC has been demonstrated to stimulate the synthesis of GSH in discussion, re-inforces the necessity of determining the GSH or total thiol groups in larvae.

·

Basic reporting

The manuscript by Benvenutti et al reports findings that N-acetylcysteine, a drug with a polypharmacological profile, blocks the motor effects of 6-OHDA in zebrafish larvae, a putative model for Parkinson's disease. The text is clear and unambiguous, with good and professional English, and professional structure of the article; the raw data was shared (although in a format that precludes automated analyses; CSV should be used, and columns organized as such). The article appears to be self-contained, but the hypothesis is generic and not directly tested.
The manuscript partially follows ARRIVE guidelines; the abstract and introduction are OK, but there is a need for more minute descriptions of the hypothesis and objectives. As it stands, the hypothesis (“we hypothesized that NAC may be a strong candidate for the prevention of PD”, lines 95-96) is generic, and not directly tested. The authors state that NAC could be a good candidate for repurposing due to the role of oxidative stress and neuroinflammation in Parkinson’s disease, but these endpoints have not been analyzed in their model.
A secondary issue is the lack of clear definition of primary and secondary outcomes assessed (also a recommendation from the ARRIVE guidelines). From the literature (e.g., 10.1016/j.neuro.2016.11.006; 10.1002/pmic.201500291; 10.1016/j.nbd.2010.06.001), we should expect motor alterations to be more relevant to modeling PD in zebrafish than, say, the morphological variables. What behavioral endpoint (from either the first or second behavioral assays) the authors consider the primary outcome? (This is also relevant for sample size calculations [see below], which should be made considering the primary outcome).
Third, the authors do not report why unequal samples sizes were used for each group. Incidentally, the group with the smaller sample size (and below we will discuss that this is not and independent replication) is the one treated with 6-OHDA, suggesting adverse events and/or censored data. If that was the case, what was the rationale for removing the data point? Did treatment results in death of one animal from this group? Were any gross anatomical alterations found in one animal? Were the data produced an outlier? If so, how was “outlier” defined?
A minor problem is that the authors did not report whether the statistical analysis used weighted or unweighted means analysis (they used unweighted-means analysis, the only way to reach the values found in Table 1). This is crucial for reproducibility.

Experimental design

The experimental design proposed by the authors is interesting in that animals were tested in both behavioral assays, therefore reducing the overall number of individuals used. There are some important problems with the design that need to be addressed.
-Your most important issue is that of randomization and blinding. Masca et al. (2015; doi:10.7554/eLife.05519), examining the problems of reproducibility in biomedical research, observed that one of the main reasons for irreproducible research is the lack of blinding/masking and convenience samples. From the text, it is not possible to know whether or not these crucial steps were taken. Were interventions (NAC) and conditions (6-OHDA) allocated at random to animals? If so, what was the method of random allocation? If not, why? Are there any other possible confounders (e.g., dishes) to which the units may need to be randomly allocated? Were care providers (lab technicians or anyone who cares for the animals) blind to treatment? Were experimenters blind to treatment? Were data analysts blind to treatment? If not, why? How was allocation concealed, and how was blinding maintained? Under what circumstances will the data be
unblinded? These questions need to be addressed, and clearly stated in the methods section.
-There is an important issue of pseudo-replication. From lines 123-127, it appears that animals were exposed as a group to each treatment (that is, animals from the 6-OHDA/NAC group were exposed to these treatments in a group of 8-11, animals from the H2O/NAC group were exposed as a group, and so forth). From the definitions and recommendations made by Lazic (2010; doi: 10.1186/1471-2202-11-5), that is a hierarchical/nested design, with a high risk for pseudoreplication in that the experimental unit is not the individual larva, but the treatment well. In zebrafish research, it is common for larvae to be treated individually in microplate wells to circumvent this problem. Please explain why the authors chose a different design, and clearly state it in the methods section.
-How was sample size calculated? Was it based on a pilot experiment to determine effect sizes? How were calculations performed? It seems that the sample sizes used for each group are small, leading to an underpowered study that inflates both Type I and Type II statistical errors; describing how the authors reached these n would be helpful in avoiding this conclusion. A good introduction to sample size calculation is Prajapati et al. (2010; http://tiny.cc/i0s6qy).
-Why did the authors choose the specific morphological endpoints they used? Using total length is, although very coarse-grained, acceptable from a developmental toxicology point of view; however, what information can be gained from analyzing head length? For that matter, what information can be gained from analyzing forebrain and midbrain width?
-A minor point is the use of the same animal in both behavioral assays. I was not able to find anything in the manuscript that affirms that, but it stands to reason, given reported sample sizes. If that was the case, it should be clearly reported, and Figure 1 should be changed to reflect that. Moreover, the authors should state whether all animals followed the same test order, or whether they were randomly distributed between locomotor first and optomotor second (and vice versa), with balancing.
-Finally, while the raw data was shared, it appears to simply be an Excel file in which values from GraphPad were pasted. International standards were not used, as commas separate decimals instead of points. Data are organized in a way that precludes quick and automated analyses (i.e., interested parties would need to copy and paste results manually, instead of automatically importing results, to programs other than GraphPad). While I strongly commend the authors for their attempt to share raw data, therefore increasing reproducibility and rigor, the file needs to be changed to a two files - one with behavioral data, and other with morphological data -, in CSV format, organized with one column per endpoint, and one line per subject.

Validity of the findings

Even though there are problems with the experimental design or its reporting, the validity of the findings is good. For example, it seems clear that 6-OHDA produced a general locomotor impairment (decreased total distance, mean speed, and maximal acceleration, and increased absolute turn angle and immobility time; decreased responsivity to an optomotor stimuli) and produced gross anatomical alterations (slightly decreased head length, decreased total length [which I infer from Figure 4F was caused by the appearance of a spinal curvature]) with no gross neuroanatomical changes; and that NAC blocked these effects, while at the same time producing gross neuroanatomical alterations (increasing forebrain width, eye distance, and total length; these later results are reported in Table 2, but the authors affirm the opposite in the text [line 194]). The small sample sizes suggest low statistical power, which could result in inflation of effect sizes and in the false discovery rate. Therefore, while valid, the results are very preliminary, and should be labeled as so, awaiting confirmatory research. The results from the neuroanatomical analyses are very coarse-grained, and appear to be not very informative (as a matter of fact, the authors sparsely discuss them) on either the neuropathology of PD, the specific effects of 6-OHDA, or the proposed neuroprotective effect of NAC; while they should be reported, the limitation of the analysis should be clearly discussed in the text.

Additional comments

As a final comment, the authors need to deeply review their Discussion section. Main issues follow:
-The language from the conclusion is boisterous, and does not reflect the actual findings. We cannot judge whether NAC exerted a “powerful neuroprotective effect” because effect sizes were not reported (and are likely to be inflated due to low statistical power). While NAC does appear to have polypharmacological profile, this was not actually tested in the experiments; and the authors did not study PD, but an animal model, which provides only weak evidence for clinical efficacy; finally, NAC itself produced gross neuroanatomical alterations (which the authors did not address); therefore, sentences such as “NAC is a strong candidate for the prevention and treatment of PD, acting in different aspects of the disease” are inaccurate and represent improper spin.
-There are many findings in the literature reporting neuroprotective effects of drugs (e.g., McKinley et al., 2005, doi: 10.1016/j.molbrainres.2005.08.014), antioxidants (e.g., Chong et al., 2013, doi: 10.1016/j.neulet.2013.02.069; Zhang et al., 2010, doi: 10.3892/ijmm.2010.571), or plant extracts (e.g., Zhang et al., 2011, doi: 10.1007/s10571-011-9731-0) on 6-OHDA-treated zebrafish. NAC was also shown to protect DAergic neurons from 6-OHDA neurotoxicity (Muñoz et al., 2004, doi: 10.1002/jnr.20107). The findings from the authors should be critically discussed in relation to this literature. Moreover, findings from the mammalian literature should also be appreciated (e.g., Zbarsky et al., 2009, doi: 10.1080/10715760500233113), as well as clinical trials on antioxidants (e.g., Snow et al., 2010, doi: 10.1002/mds.23148) and systematic reviews on anti-inflammatory drugs for PD patients (Rees et al., 2011, doi: 10.1002/14651858.CD008454.pub2); these will help put the results in context.

Ethical approval statement – OK

Necessity and ethics of experiments – OK

Animal research policies – Some aspects of the ARRIVE guidelines were not followed; critically, randomization/blinding was not reported, nor was the rationale for the small sample sizes.

---

## Round 0.2 · accepted · Accept

I have received the reviewer's report for your revised manuscript and I have myself looked at your manuscript. Thank you for correcting it and taking into consideration our queries.

# ·

Basic reporting

No comment

Experimental design

No comment

Validity of the findings

No comment

Additional comments

The present version of the manuscript greatly improved its reporting in terms of transparency and rigor.